# Quality in Psychiatric Care in the Community Mental Health Setting from the Perspective of Patients and Staff

**DOI:** 10.3390/ijerph20054043

**Published:** 2023-02-24

**Authors:** Juan Francisco Roldán-Merino, Manuel Tomás-Jiménez, Agneta Schröder, Lars-Olov Lundqvist, Montserrat Puig-Llobet, Antonio R. Moreno-Poyato, Marta Domínguez del Campo, Sara Sanchez-Balcells, Maria Teresa Lluch-Canut

**Affiliations:** 1Mental Health Department, Campus Docent Sant Joan de Déu Private Foundation, University of Barcelona, C/Sant Benito Menni, 18-20, 08830 Sant Boi de Llobregat, Spain; 2Grupo DAFNiS, Campus Docent Sant Joan de Déu, Universitat de Barcelona, 08830 Sant Boi de Llobregat, Spain; 3Mental Health, Psychosocial and Complex Nursing Care Research Group—NURSEARCH, University of Barcelona, 08907 Barcelona, Spain; 4Parc Sanitari Sant Joan de Déu, 08830 Sant Boi de Llobregat, Spain; 5Patient Safety Research Group, Hospital Parc Sanitari Sant Joan de Déu, 08830 Sant Boi de Llobregat, Spain; 6University Health Care Research Center, Faculty of Medicine and Health, Örebro University, 701 85 Örebro, Sweden; 7Department of Nursing, Faculty of Health Care and Nursing, Norwegian University of Science and Technology (NTNU), 2815 Gjövik, Norway; 8Public Health, Mental Health and Maternal Infant Nursing Department, Nursing College, University of Barcelona, Health Sciences Campus Bellvitge, Hospitalet de Llobregat, 08907 Barcelona, Spain; 9Etiopatogenia I Tractament Dels Trastorns Mental Severs (MERITT), Institut de Recerca Sant Joan de Déu, Santa Rosa 39-57, 08950 Esplugues de Llobregat, Spain

**Keywords:** quality of care, community care, psychiatric care, patients’ perspective, staff’s perspective

## Abstract

The current paradigm of mental health care focuses on care provided in the community, increasingly moving away from hospital care models that involve considerable economic burden. Patient and staff perspectives on the quality of psychiatric care can highlight strengths and areas for improvement to ensure better care provision. The aim of this study was to describe and compare perceptions of quality of care among patients and staff in community mental health services and to determine possible relationships between these perceptions and other study variables. A comparative cross-sectional descriptive study was conducted in a sample of 200 patients and 260 staff from community psychiatric care services in the area of Barcelona (Spain). The results showed high overall levels of quality of care from patient (m = 104.35 ± 13.57) and staff (m =102.06 ± 8.80) perspectives. Patients and staff both gave high scores to Encounter and Support factors, while factors concerning patient Participation and Environment received the lowest scores. Continuous assessment of the quality of psychiatric care in the community setting is essential to ensure the highest quality of care, taking the perspectives of those involved into account.

## 1. Introduction

In recent decades, the World Health Organization has instituted a paradigm shift in mental health care at international level, which involves a move away from institutional care models in favor of community care provision [1]. The aim is to carefully balance mental health care between community and hospital services while prioritizing the relocation of the majority of care provision so that it is near to or at people’s homes [2]. Inadequate community psychiatric care continuity can lead to referrals to long-stay units and institutionalization, which entail increases in health costs and reduced quality of care [3,4]. As such, community care systems should be strengthened to deal with patients with differing degrees of healthcare needs and professional training should include a focus on community care as a central component [5].

Patient satisfaction is an important aspect in the assessment of health services [6] and it has been shown that coordinated care provided in the community is associated with the perception of greater satisfaction among mental health patients [7]. However, patient satisfaction is not necessarily synonymous with quality [8], as measurement of satisfaction in terms of service indicators is not the same as exploring patients’ definitions of care quality and which factors best facilitate their recovery [9]. The main difference between satisfaction and quality of care is that the concept of quality of care should include the perspectives of all parties involved [10]. Nevertheless, assessment of care quality is usually conducted from the staff’s point of view [11], even though it has been observed that the perceptions of patients and staff do not coincide when it comes to defining what constitutes high quality care [12].

Patients have the right to receive care that corresponds to their needs [13] and the services provided by community mental health staff should be considered acceptable and accessible by patients and their families [2,13]. This includes intensive case management, early crisis interventions and rehabilitation services [13]. Indeed, quality of care received is an influential aspect in patients’ treatment outcomes [14].

It is crucial to examine patients’ perceptions of quality of care received [15,16] and, in fact, patients’ perception measures are judged to be of greater value than even standard measures of quality of care such as clinician or referrer indicators [17]. Working in tandem with mental health service users is considered an essential value recognized in various countries [18]. In particular, users’ active participation can be implemented at different levels that can range from a more individual approach, in which the patient takes part in decision-making related to treatment and care received [13], to their inclusion in service planning and assessment, in addition to research activities [19]. There are even authors who defend patient participation in the design of instruments to measure quality of care, highlighting the importance of their experiences and points of view, which can be a source of valuable information [20,21].

One of the most important indicators in the field of health in Catalonia is the perception, experience and satisfaction survey plan for users of the Catalan Health Service. The most recent results for community mental health centers from 2021 showed that degree of satisfaction (7.82 out of 10) and adherence (81% intention to return to the center) had dropped slightly compared with results from the previous study in 2018 (satisfaction 7.92/10 and adherence 82.9%). The aspects rated most highly by patients were the treatment received from professionals and the cleanliness of the center, while the aspects with the lowest rating were contact with the center and frequency of visits. An important limitation of this study is that it only includes the patients’ point of view while it would be valuable to also explore that of professionals [22].

Mental health staff define quality of care as a multidimensional concept, which is positive and normative [23]. Understanding the staff’s perspectives with respect to quality of care provided facilitates identification of areas for improvement and promotes the development of new strategies that will subsequently benefit service users [24,25]. It is important to emphasize good coordination and inter-service relationships, as well as solid support for professionals from organizations, elements which help to avoid exhaustion and burn-out and thus contribute to better patient care [26].

Policies have been developed worldwide to encourage more personalized, efficacious care planning, although evidence suggests that these innovations have not met all their goals and further research is needed [27]. The use of robust, validated instruments can identify scope for improvement in care provision and the quality of mental health services [28]. To ensure the best possible quality of care for people with mental illness, it is important to conduct a continuous process of assessment and improvement [29,30].

The aim of the present study was to describe and compare perceptions of quality of care from the perspective of both patients and staff in community mental health services and to determine possible relationships between these perceptions and other study variables.

## 2. Materials and Methods

### 2.1. Design

Comparative cross-sectional descriptive study. Mean values of perceived quality of care were determined in two study samples at a single timepoint for later comparison.

### 2.2. Participants

The study was conducted using two samples; one of patients and one of staff. Both samples were recruited from community mental health facilities from Parc Sanitari Sant Joan de Déu in 1 district from Barcelona city and 9 nearby towns. This institution offers a concerted service specializing in mental health to the regional health system that is fully public and free of charge to users.

#### 2.2.1. Patients

A total of 200 patients treated at distinct community mental health facilities in the Barcelona area were recruited through consecutive non-probabilistic sampling. Inclusion criteria established were being older than 18 years, having a diagnosis of mental disorder, being in follow-up at a community mental health facility at the time of the study and agreeing to participate voluntarily. Exclusion criteria were inability to understand or express oneself in Spanish, severe cognitive impairment, and organic disorder and/or intoxication due to drug use at the time of assessment. Data collection was conducted between February 2020 and March 2022.

#### 2.2.2. Staff

The sample of staff consisted of 260 staff from a number of disciplines at different community mental health facilities in the Barcelona area, recruited through non-probabilistic convenience sampling. Inclusion criteria were being actively employed in the aforementioned facilities and voluntarily agreeing to participate. The sole exclusion criterion was having less than one year of experience in the area of mental health. Conditions for the work environment at the centers studied follow national health system guidelines. Data collection was conducted between February 2019 and February 2020.

### 2.3. Variables and Sources of Information

The present study used the instruments Quality in Psychiatric Care-Outpatient (QPC-OP) for patients and Quality in Psychiatric Care-Outpatient Staff (QPC-OPS) [31] for staff. Using these instruments, patients and staff respond to the same questions about the quality of care received or provided.

Both instruments had previously been validated in Spanish and showed good psychometric properties in terms of reliability and construct validity [32,33]. The instruments are based on the definition of quality of care developed by Schröder et al. [34,35] following a phenomenographic study in psychiatric patients [36].

Each instrument consists of a total of 30 items distributed across 8 factors as follows: F1 Encounter (6 items) represents aspects covering the staff-patient interpersonal relationship where the level of respect, listening, empathy and staff concern for patients are assessed. F2 Participation-Empowerment (3 items) and F3 Participation-Information (5 items) reflect the level of involvement patients have in their care as well as whether the information received is sufficient for them to make decisions about their care. F4 Discharge (3 items), on the other hand, deals with continuity of care at the facility in question, while F5 Support (4 items) reflects support received by patients from staff regarding the stigma associated with mental illness. F6 Environment assesses perception of the degree of safety at the center. F7 Next of kin represents the degree of participation and respect offered to patients’ relatives, while F8 Accessibility (4 items) evaluates contact with the center and the staff assigned to the patient. Each item begins with the phrase “I experience that…” and is scored on a Likert-type scale with four response options ranging from 1 (completely disagree) to 4 (completely agree) with a further ‘not applicable’ option for each one if required. Scores can be obtained globally or by factors; the maximum global score is 120 points and the minimum 30 points, such that a high score in each factor or in the overall instrument indicates a good perception of quality of psychiatric care. On the other hand, a low score implies a need for an improvement intervention in the identified areas.

In addition, data on the following variables were collected in each group: age, sex, nationality, perceived mental health and perceived physical health.

In addition, collected in the patient group were the following: level of shared living (living alone/living with others), educational level, time in contact with mental health services, participation in care planning, awareness of claims channels and number of visits in the previous year.

In addition, collected from the staff group were data on professional categories and number of years worked at the center.

### 2.4. Data Collection Procedure

Both patients and staff were invited to take part by contact personnel who informed them about study procedures and aims. All participants were informed verbally about the purpose of the study in a manner adapted to their understanding, ensuring that they fully comprehended all the information provided. Subsequently, an information sheet and informed consent document were delivered, which were signed to confirm their completely voluntary participation. It was made clear that declining to participate would have no effect on their care (in the case of patients) or on their professional situation (in the case of staff). Each participant was informed of their right to withdraw from the study at any time if they so wished. Figure 1 shows the flow diagram for the data collection process.

The study was approved by the research ethics committee at Fundació Privada per a la Recerca i la Docència Sant Joan de Déu FSJD and assigned the CEIC code PIC-83-16.

### 2.5. Data Analysis

All statistical analyses were performed using SPSS version 28, and a descriptive analysis of the database was conducted, generating descriptive statistics for numerical variables and frequency tables for categorical variables.

To allow comparison of the factor scores from the different instruments and factors, the mean score of each factor was divided by the number of items it contained. The Student’s *t*-test was used to compare the total and between factor scores between patients and staff.

To study the relationship with categorical sociodemographic variables with 2 possible categories, the *t* test or Mann–Whitney U test were used, depending on whether normality could be assumed or not in the distribution of the variable. Finally, to study the relationship with sociodemographic variables with more than 2 categories, ANOVA or Kruskal–Wallis tests were used, depending on whether it could be considered that there was normality in the distribution of the variables. To study the relationship between the scale factor scores (or the total score) with numerical sociodemographic variables, the Pearson correlation was calculated.

## 3. Results

### 3.1. Patients

#### 3.1.1. Description of the Sample

The patient sample consisted of 200 participants. Some 28.4% lived alone compared with 71.6% who lived with others. The participants assessed had been in contact with mental health services for 11.52 ± 11.11 years. A total of 78.4% of participants stated that they had taken part in their care planning, while the remaining 21.6% said that they had not. Regarding educational level, 11.8% did not complete primary education, 21% completed primary education, 17.9% secondary education, 32.8% professional training-high school, and 16.4% higher or university education.

At the time of the survey, 6.3% of participants stated that they had not visited the outpatient center on any occasion during the previous year, while 13.7% said that they had visited once, 16.8% between 2–5 times, 20% between 6–10 times and the remaining 43.2% had made more than 10 visits to the center in the previous 12 months. In all, 51% of participants declared that they were aware of the claims channels, in the case that they were not satisfied with their treatment, while the remaining 49% stated that they were not aware of them. Table 1 shows the sociodemographic variables of both study groups expressed as mean and standard deviation or frequency and percentage according to type of variable.

#### 3.1.2. Perception of Quality of Care

Table 2 shows the score values for each factor, revealing that the highest scores given by the patients were for the factors F1 Encounter (3.59 ± 0.57) and F5 Support (3.59 ± 0.57), while the lowest scores were for the factors F3 Participation-information (3.33 ± 0.61) and F2 Participation-empowerment (3.22 ± 0.64). A great number of patients (95%) rated the quality of care received as high, bearing in mind the cut-off point established by the original authors of a mean global score greater than 2.5 points calculated for the 30 items [31].

A weak positive correlation was observed between age and the global instrument score (r = 0.183, *p* = 0.011). Those patients who stated having participated in care planning had higher scores (m = 105.06 ± 15.08) in the perception of quality of care than those that did not (m = 99.97 ± 14.74, *p* = 0.016).

The results of the Kruskal–Wallis test revealed statistically significant differences between the QPC-total score and the number of visits made to the center in the previous year (*p* < 0.001). The post-hoc analysis showed that those who had visited more than 10 times in that year had higher mean scores in perception of quality of care (m = 109.80 ± 9.84) than those who had visited 2–5 times (m = 99.21 ± 14.13, *p* < 0.001), 1 time (m= 96.92 ± 17.03, *p* < 0.001) or had not visited (m = 96.25 ± 18.14, *p* = 0.003).

The Kruskal–Wallis test showed statistically significant differences between global quality scores and perceived mental (*p* < 0.001) and physical health (*p* = 0.011). With regard to perception of mental health, the post-hoc analysis showed that patients who reported very good mental health had higher scores (m = 111.38 ± 17.74) than those who referred to good mental health (m = 106.40 ± 11.30, *p* = 0.002), neither good nor bad (m = 102.03 ± 13.21, *p* < 0.001) and bad mental health (m = 96.68 ± 16.89, *p* < 0.001). On the other hand, the scores of those who reported good mental health (m = 106.40 ± 11.30) were higher than those who had bad mental health (m = 96.68 ± 16.89, *p* = 0.005). Regarding perception of physical health, the post-hoc analysis only showed, with statistically significant results, that people who reported having very good physical health had higher mean quality of care scores (m = 109 ± 23.27) than those who reported bad physical health (m = 99.03 ± 16.25, *p* =0.002).

No statistically significant results were obtained that related global instrument scores and sex (*p* = 0.553), nationality (*p* = 0.752), level of shared living (living alone/living with others) (*p* = 0.559), educational level (*p* = 0.180), time in contact with mental health services (r = 0.24, *p* = 0.75) and being aware of the claims channels if they were dissatisfied with their treatment (*p* = 0.194).

Table 2 shows the mean and standard deviation values for the different QPC-OP/OPS instrument factors, along with their total value.

### 3.2. Staff

#### 3.2.1. Description of the Sample

The sample consisted of a range of professional categories distributed as follows: 29.61% nurses, 20.38% psychiatrists, 16.53% psychologists, 8.84% social educators, 7.7% case managers, 7.30% social workers, 5% administration and 4.61% occupational therapists, with a mean of 8.68 ± 7.60 years worked at the center. Table 1 shows other variables of interest.

#### 3.2.2. Perception of the Quality of Care

Table 2 shows the score values for each factor, showing that the highest scores were for the factors F1 Encounter (3.68 ± 0.37) and F5 Support (3.64 ± 0.41), while the lowest scores were for the factors F2 Participation-empowerment (3.10 ± 0.45) and F6 Environment (2.99 ± 0.53). All staff members bar one rated the quality of care provided as high, bearing in mind the cut-off point established by the original authors of a mean global score greater than 2.5 points calculated for the 30 items [31].

Statistically significant differences were detected with respect to nationality and global score, where the scores of the Spanish staff (m = 102.17 ± 8.97) were higher than those of other ethnic backgrounds (m = 94.90 ± 8.70) (*p* = 0.009). 

No statistically significant correlations were detected between global score and age (r = 0.07, *p* = 0.240), sex (*p* = 0.674), time worked at the center (r = 0.096, *p* = 0.123), or perceived mental (*p* = 0.562) or physical health (*p* = 0.351).

Statistically significant differences were found when comparing the global scores between patients and staff (*t* = 2.06; *p* = 0.040), with patients showing higher scores (m = 104.35 ± 13.57) than staff (m = 102.06 ± 8.80). Statistically significant differences were also observed in three of the eight factors that make up the questionnaire, with patients showing higher scores in all three.

## 4. Discussion

The aim of this study was to describe and compare the perception of quality of care from the perspectives of patients and staff in community mental health services and to determine possible relationships between these perceptions and other study variables.

In general, the global quality of care scores given by both patients and staff were high. Spanish community mental health patients presented global quality scores which were higher than those of other countries in the community setting [5,31,37], and also higher than in other contexts such as hospital [5,34] and forensic settings [38,39] using instruments from the same family. Similarly, Spanish staff had global scores which were somewhat higher than staff in other countries in community [40], hospital [41] and forensic [42,43] contexts with instruments from the same family.

It should be mentioned that patients gave higher scores in quality of care received than staff. This is in contrast to studies from other fields in which staff rated the quality of care more highly than patients [44,45], although a possible explanation for this may be that staff tended to overestimate the quality of care provided [46]. In previous research in both the physical and psychiatric areas, the differences between patients and staff can be associated with discrepancies when defining a particular concept [47]. However, the dissimilarities in points of view between patients and staff may also be due to differences in experiences, knowledge, expectations and/or educational level [44,48].

In the case of patients, the factors with the highest scores were F1 Encounter and F5 Support. This finding is consistent with other studies using QPC instruments where encounter was the most highly [5,31,34,37] or second most highly rated factor [5], and support was the second most highly rated factor by patients [5,38,39]. On the other hand, those factors which received the lowest scores were F3 Participation-information and F2 Participation-empowerment. In similar studies, factors referring to the participation of patients in their own care received lower scores [34,38,39,49] as well as those covering aspects related to information and participation [50,51].

Staff gave the highest scores to the factors F1 Encounter and F5 Support. In studies using instruments from the same battery, staff gave encounter the highest [40] or second highest score [41,52], while support also ranked first [41,52] or second [40,42,43] among the factors. In contrast, F2 Participation-empowerment and F6 Environment obtained the lowest scores. In the previously mentioned studies, factors related to participation were rated second to last [42] and the factor environment, which was closely associated with perceived safety by staff, occupied the final positions in the hospital [41,52], community [40] and, in particular, the penitentiary setting [42,43]. This feeling of a lack of safety reported by staff may be due to episodes of violence (mainly verbal and occasionally physical) that can occur in the psychiatric setting [53]. Nurses should apply scientific evidence in practice to improve quality of care and safety in the center [54], taking into account that the safety of both patients and staff is an important aspect of quality of care [53].

It should be kept in mind that the factor F1 Encounter contains items reflecting the interpersonal relationship between staff and patient. Staff-patient interaction has been identified by some authors as a central element in quality of care [16,55] and should be considered a priority by staff in ensuring high care quality [56]. From the patients’ perspective, the importance of contact and the interpersonal relationship has been emphasized, where staff can dedicate time to listening and exploring patients’ feelings rather than mainly focusing on their daily routines and tasks [57], especially as this can influence whether or not the patient returns to the center [58].

Although the results of this study regarding F5 Support scores represent good progress in the fight against the stigma around mental health, there is an urgent and ongoing need for recovery-centered anti-stigma education addressed to health staff. This would be of benefit to both hospitals and outpatient units in helping to acquire tools to tackle public and internalized stigma associated with mental illness [59]. Beckers et al. [60] reported a certain preference among patients for community mental health treatment as it is associated with less stigma than older mental health services, bearing in mind that the experience of stigma affects the patient, their relatives and others outside the family [61].

Concerning the great importance of information in care provision, it is vital to adapt the information to the patients in terms of their unique living situation and previous experiences of care [62,63]. Previous studies confirm that patients who consider that they are well-informed rate the quality of care received more highly than those who are less well-informed, an aspect that staff should take into account [31,34,38,39]. Being aware of the claims channels could be considered as a piece of information that should be available to the patient even though it may appear contradictory that the present study did not identify this awareness as a facet that influenced perception of quality of care. In fact, this finding is not consistent with previous similar studies which reported that patients who were aware of the claims channels perceived better quality of care [38,64,65].

Patients who participated in their own care planning gave higher scores in quality of care received and these data are important considering that the factors covering participation were rated lowest by patients and F2 Participation-empowerment was rated lowest by staff. Crawford et al. (2002) conducted a review in which they found that patients involved in their care attained better health, were more satisfied and enjoyed better quality of life but also remarked that the effects of their participation in the quality and effectiveness of the services is still unknown [66]. It is interesting to reflect on the low score given to participation by both groups considering the importance given to the participation of patients in the redesign of health processes [67]. However, the low scores associated with participation appear to be a frequent issue in the psychiatric setting [39] where greater patient autonomy and involvement in decision-making should be encouraged [49]. The results, in this sense, may reveal relatively more patient dissatisfaction with their opportunities to take part in their care and a lack of satisfaction among the staff themselves at not being able to offer patients adequate options for involvement [44], which could partly be influenced by a conceptual difference between staff and patients on what constitutes participation in care [68].

A weak positive correlation was observed between age and global instrument score, suggesting that the older the patient, the higher the quality score given. This finding is in line with previous studies that indicate that older people are usually less critical than younger ones when rating quality of care received [34,64,69], which may be due to older people having lower expectations in this regard [30] or even that staff treatment of older people is more attentive and respectful [70]. This relationship was not observed for staff, where age was not an influential factor in the perception of quality of care provided. This finding is in contrast to previous studies in which older staff perceived better quality of care offered than younger staff [42].

With regard to sex, no relationship was found with quality of care in either group. This differs from the findings of previous studies that found higher quality of care scores among female patients than males [50,71]. On the other hand, there are other studies that observed exactly the opposite, that is, that male patients gave higher scores in quality of care than females [38,49,72,73]. With regard to staff, nor was any relationship found between perception of quality of care and sex in other studies [42].

It should, nevertheless, be pointed out that previous research in sex and age did not show a clear association with satisfaction or quality of care received by the patient [35,74,75].

Similarly, in the case of patients, nationality was not a factor that influenced perception of quality of care, which is consistent with previous studies [38,75], although Spanish staff perceived better quality of care provided than staff from other ethnic backgrounds—a finding not observed in previous studies where the nationality of staff appeared to have no bearing on the perception of quality [42,44].

Living alone or with others was not a factor which had an impact on perception of quality among patients, which has also been reported in similar studies in other settings [38], although a study in a hospital setting did find that living alone was a factor underlying lower scores in discharge [75].

Educational level was not a factor that affected perception of quality of care among patients either, which is in keeping with previous research that also found no association between these variables [38,75]. Some studies reported that patients with a low educational level tended to give lower scores for quality of care [64]. However, in contrast, other studies indicated that patients with a higher educational level perceived poorer quality of care than those with a lower level of education [30,65,76], and one explanation could be that people with a higher educational level are more demanding of the services they receive [77] and they may rate care quality as poorer if their expectations are not met [65].

In the case of patients, time in contact with mental health services was not a factor that impacted their perception of quality of care. With regard to staff, years of experience in the mental health area did not affect their perception of quality, which is in line with the findings of other similar studies [44].

Both study groups were asked about their perception of physical and mental health at the time of the interview. In patients, a relationship was observed between these health perceptions and their assessment of quality of care received, such that people who reported having good mental health gave a higher score in perception of quality of care. This association between perceived mental health and perceived quality of care in patients was previously described in similar studies [5,38,65]. Likewise, patients who reported good physical health gave higher scores to quality of care received and this influence of health status on the perception of quality of care was also previously observed [49,65,78]. The literature describes how quality psychiatric care leads to better health outcomes [79] although, as ours is a cross-sectional study, it is not possible to determine whether high quality care produces better health in the patients treated or better health influences patient perception of the facility attended—a limitation already found in a similar study to ours [65]. In the case of staff, their perceptions of physical and mental health were not factors which affected their perception of quality of care provided.

It is essential to explore patients’ experiences and perceptions regarding satisfaction and quality of services as these are important indicators of their intention to visit the center again [5,80,81,82], adherence to treatment and enjoyment of good quality of life [82], and this knowledge can contribute to ensuring continuity of quality care. Consideration of the perspectives of patients and staff is of great value when implementing organizational changes [45,48] and regular assessments, provision of adequate resources, training, leadership and professional support greatly increase the opportunities for work improvements [83].

The results of this study should be interpreted with caution due to a number of limitations. The main limitation is that as a comparative cross-sectional study, no follow-up assessments were conducted and so, it was not possible to assess any type of progress. The study design also prevented us from assessing the influence of quality of care on the physical or mental health of those treated and future studies should explore this issue. A further limitation concerns the data-collection periods for each study group, which were not simultaneous, and some factors may have arisen between the two series of dates which affected the perception of quality in either or both groups. As such, it is recommended that in future studies data collection periods should be concurrent. It should also be noted that there are fewer people in the patient group than in the staff group and future research should ensure equivalence in group sizes. Data on severity of mental health illness was not collected in this study and its inclusion as a study variable should be considered in future research to determine whether it is a factor influencing perception of care quality. We would recommend the addition of an open-ended questionnaire in future studies similar to this one to gather participants’ suggestions regarding quality of care in the center studied.

The findings of this study have a series of implications for clinical management. Aspects covering the staff-patient relationship and anti-stigma attitudes are of great importance to both staff and patients while the issue of patient participation in their care process remains unresolved. Thus, it is essential to develop patient participation in a way that is satisfactory to both parties involved. On the other hand, it is vital to consider the patient’s profile and adapt care to characteristics such as age so that it is adjusted to care recipients’ individual needs and their environment.

## 5. Conclusions

The results of our study show that patients rated the quality of psychiatric care more highly than staff and this may be due to the demands that staff placed on themselves at work. It was observed that F1 Encounter and F5 Support scores received were strengths in the context of care received while participation emerged as the main area for improvement in the view of both patients and staff.

The results of this study highlight the importance of an interactive role for patients and staff, working together as a team to ensure quality in psychiatric care. Systematic, continuous assessments of care quality of care are vital in guaranteeing quality services that can adapt to any obstacles that may arise, while treating patients in their community can help to avoid costly hospital admissions. The dual perspective contributes valuable information, both from staff as care providers and strategists in care improvements, and patients due to their unique point of view according to their living situation and, in particular, the largely voluntary nature of their connection to the community facilities where they are treated.

## Figures and Tables

**Figure 1 ijerph-20-04043-f001:**
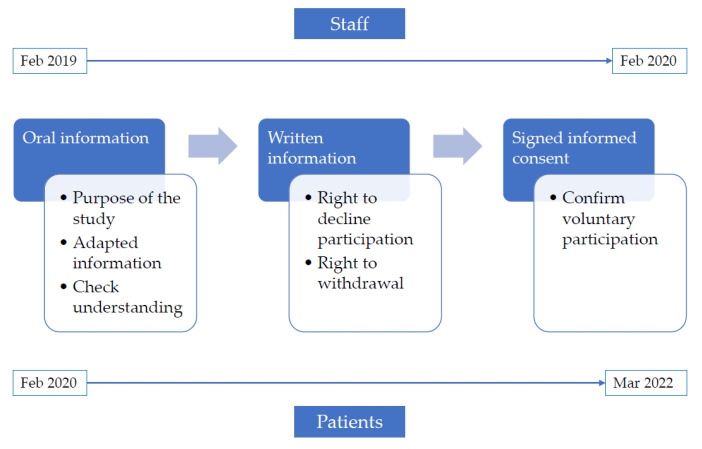
Data collection flow diagram.

**Table 1 ijerph-20-04043-t001:** Study groups’ sociodemographic characteristics.

Background Variable	Patients. n (%)	Staff. n (%)
n	200	260
Sex		
Woman	99 (49.5%)	192 (73.8%)
Man	100 (50%)	67 (25.8%)
Missing data	1 (0.5%)	1 (0.4%)
Age M (SD)	46.15 (13.74)	40 ± 10.03
Nationality		
Spanish	184 (92%)	248 (95.4%)
Other	16 (8%)	11 (4.2%)
Missing data	0 (0)	1 (0.4%)
Perceived physical health		
Very bad	16 (8%)	0 (0)
Bad	51 (25.5%)	7 (2.7%)
Neither good nor bad	59 (29.5%)	37 (14.2%)
Good	58 (29%)	176 (67.7%)
Very good	14 (7%)	38 (14.6%)
Missing data	2 (1%)	2 (0.8%)
Perceived mental health		
Very bad	7 (3.5%)	0 (0)
Bad	35 (17.5%)	1 (0.4%)
Neither good nor bad	61 (30.5%)	22 (8.5%
Good	69 (34.5%)	174 (66.9%)
Very good	26 (13%)	61 (23.5%)
Missing data	2 (1%)	2 (0.8%)

**Table 2 ijerph-20-04043-t002:** Means (M) and Standard Deviations (SD) of Perceived Quality of Outpatient Care Among Patients and Staff.

Factors	Patients Mean (SD)	Staff Mean (SD)	*t*	*p*
F1. Encounter	21.74 (3.08)	22.14 (2.19)	−1.532	0.126
6 items	3.59 (0.57)	3.68 (0.37)		
F2. Participation-Empowerment	9.74 (1.85)	9.33 (1.64)	2.432	0.015 *
3 items	3.22 (0.64)	3.10 (0.45)		
F3. Participation-Information	16.80 (2.83)	16.81 (2.34)	−0.046	0.963
5 items	3.33 (0.61)	3.35 (0.57)		
F4. Discharges	10.42 (1.80)	10.40 (1.14)	0.082	0.935
3 item	3.45 (0.62)	3.46 (0.39)		
F5. Support	14.46 (2.07)	14.60 (1.61)	−0.786	0.432
4 items	3.58 (0.57)	3.64 (0.41)		
F6. Environment	10.55 (1.73)	8.98 (1.61)	9.852	0.0001 *
3 items	3.49 (0.62)	2.99 (0.53)		
F7. Next of kin	7.02 (1.19)	7.08 (0.87)	−0.604	0.546
2 items	3.48 (0.63)	3.53 (0.44)		
F8. Accessibility	13.59 (2.55)	12.68 (2.44)	3.866	0.0001 *
4 items	3.37 (0.67)	3.16 (0.61)		
Total QPC score	104.35 (13.57)	102.06 (8.80)	2.06	0.040 *
30 items	3.45 (0.79)	3.39 (0.69)		

SD—Standard deviation, *t*—t-Student Fisher, *p*—*p*-value. * significant at *p* < 0.05.

## Data Availability

The data presented in this study are available upon request from the corresponding author.

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
