# Peer review of "Quality in Psychiatric Care in the Community Mental Health Setting from the Perspective of Patients and Staff"

_ijerph, 2023, doi:10.3390/ijerph20054043_

Round 1
Reviewer 1 Report
Dear Authors
your paper "Quality in psychiatric care in the community mental health setting from the perspective of patients and staff" is important and well written when community services setting are noticed to be better care places for many patients than institutions according to various studies. Inspite of ill health and age, all patients seem (universally) to appreciate the quality of encounter and support given by good staff. So human factors are core issues in quality of care. I got only some minor comments on your paper:
1) were the patients and staff recruited from the SAME mental health services in community settings? What do you think is there variation between centers about the work and caring conditions?
2) In terms of staff, you should take into account the work settings, management, human resources etc which have an effect to how the care is possible to be given. This is one shortcoming in your paper.
3) How about the severity of mental health problem, how does it have an effect on patient's evaluation of quality of the care? Can you study the quality indicators in terms of severity of the mental ill health ?
4) how easy it is to access to community mental health setting in Barcelona? Does patient need an referral to be able to have care there? How the care is usually organised? Only day time? Is Food given given there and are there social activities available to the patients? Are patients mainly in disability pension or at work? Many times disability to work mean dependency on sick leave/pension which is usually low in terms of living costs, and these people for example with schitsofrenia form their 20's mean that they have no work history. This may lead to living in poverty with poor physical and mental health. So it would be probable that their care needs are larger than others in terms of having proper and safe living conditions. How about patients with alcohol problem and mental health problems?
5) when patients in community care settings are referred into hospital care?
Author Response
Dear Reviewer,
Before we address your observations point by point, the authors would like to thank you for each of your comments as they will undoubtedly help to improve the quality of our manuscript.
1) were the patients and staff recruited from the SAME mental health services in community settings? What do you think is there variation between centers about the work and caring conditions?
Thank you for allowing us to clarify this point. Both the patient and staff samples were recruited from the same community mental healthcare facilities and these facilities are part of the same institution. Thus, in this respect, the infrastructure, care model and working conditions follow the same institutional pattern. In section 2.2, we have specified that both samples have the same institutional origin.
2) In terms of staff, you should take into account the work settings, management, human resources etc which have an effect to how the care is possible to be given. This is one shortcoming in your paper.
We appreciate your pointing this out. The working environment in the centers studied is based on health care system guidelines so that, regarding ratio, working and salary conditions, it is comparable with the rest of the health system. This observation is reflected in section 2.2.2.
3) How about the severity of mental health problem, how does it have an effect on patient's evaluation of quality of the care? Can you study the quality indicators in terms of severity of the mental ill health ?
Thank you for this excellent observation. Unfortunately, we must state that data on this variable were not collected and, as such, a more in-depth analysis of the severity of the mental health issue could not be carried out. However, as your observation is highly valid, we have included it at the end of section 4 as a limitation and as a recommendation for future studies.
4) how easy it is to access to community mental health setting in Barcelona? Does patient need an referral to be able to have care there? How the care is usually organised? Only day time? Is Food given given there and are there social activities available to the patients? Are patients mainly in disability pension or at work? Many times disability to work mean dependency on sick leave/pension which is usually low in terms of living costs, and these people for example with schitsofrenia form their 20's mean that they have no work history. This may lead to living in poverty with poor physical and mental health. So it would be probable that their care needs are larger than others in terms of having proper and safe living conditions. How about patients with alcohol problem and mental health problems?
Thank you for the opportunity to clarify these points. The institution to which the mental health facilities studied belong offers a concerted service to the regional health system. Consequently, patient access to the facilities is completely public with no economic cost to the person treated. When a primary care or hospital service detects a mental health problem, the person is referred to one of the specialized services so a closer, more individualized follow-up can be conducted. This has been explained in section 2.2 to help the reader understand the context more easily. In these facilities, the patients are only cared for during the day and food is not given to them. The facilities are comparable to outpatient centers but for mental health. Thus, patients do not participate in social activities in the center itself as there are other facilities in the same network for this purpose but these did not fall within the scope of our study.
All patients receive income as active workers or as recipients of a contributory or non-contributory pension, in addition to receiving a series of social benefits according to their situation and income level. In cases where the level of income is insufficient, there are medium/long-term care facilities available with their own care teams, although these were not included in our study.
Regarding people with both addiction and mental health problems, these are treated at another regional institution specializing in the field of addictions.
5) when patients in community care settings are referred into hospital care?
Thank you. When the outpatient care teams detect a decompensation or a situation that gives cause for concern, more intensive outpatient follow-up mechanisms are activated to avoid hospital admission. However, if this follow-up proves inadequate to deal with the needs of the person in this situation, outpatient services promptly liaise with hospital services to arrange an admission of the shortest possible duration.
Reviewer 2 Report
Dear authors,
The topic is this study is very interesting because it is directly collected to community needs. This study requires more description of "why" and "what's next". Some notes and recommendations follow:
- What is the issue? Describe issues of community health care both from non-academic or academic reports. For example, country A found that patients were not satisfied due to bad management, etc. The existing issues will strengthen the “importance” value of this study.
- Did the authors decide to select participants with mental disorders only? If yes, why? Explain more if the health care is only for people with MD?
- How many cities were covered in this study?
- If It’s only one line, such as: All statistical analyses were performed using the SPSS version 28. 150, Data collection was carried out between February, 2019 and February, 2020. 110, it would be better not to put it in one paragraph format. Merge it with other sentences. A paragraph commonly consists of between 5 and 8 sentences.
- had higher scores than those 208 who had bad mental health – what is the explanation for this finding?
- This finding is in line with previous studies that indicate that older people are usually less critical than younger ones when rating the quality of care received – what is the recommendation for this? Should differentiated services based on patients’ ages be provided?
- Was there an open-ended questionnaire in this study? If not, please consider suggesting further research to use the methods to gain recommendations from participants.
Thank you! Best of luck!
Author Response
Dear authors,
The topic is this study is very interesting because it is directly collected to community needs. This study requires more description of "why" and "what's next". Some notes and recommendations follow:
Dear Reviewer,
We deeply appreciate the contributions you have made to our work as they have allowed us to substantially improve our manuscript. We would like to comment on your observation regarding the quality of English in our paper. The translation was done by a native translator with extensive experience in translating scientific articles and, if you consider it necessary, we would be more than happy to forward the relevant invoice. Nevertheless, the translator has done a thorough revision of the article improving the English level and contributed to the new additions to the article in accordance with the reviewers’ recommendations.
- What is the issue? Describe issues of community health care both from non-academic or academic reports. For example, country A found that patients were not satisfied due to bad management, etc. The existing issues will strengthen the “importance” value of this study.
We are grateful for your comments and, to underline the importance of the study, we have added a paragraph to the introduction on the most relevant study carried out regularly and systematically in Catalonia (Spain) which assessed patient satisfaction with the various health facilities, including mental health outpatient centers. The data show a slight decrease in satisfaction and adherence (understood as the intention to return to the center) compared with previous data from 2018. It should be noted that this only explored patients’ perceptions and professionals’ perceptions are not described.
- Did the authors decide to select participants with mental disorders only? If yes, why? Explain more if the health care is only for people with MD?
The services studied are outpatient centers specializing in mental health. This is clarified in section 2.2 to avoid any misunderstanding.
- How many cities were covered in this study?
In fact, this study covers ten areas in and around Barcelona city. Our apologies for any confusion. This information has been included in section 2.2.
- If It’s only one line, such as: All statistical analyses were performed using the SPSS version 28. 150, Data collection was carried out between February, 2019 and February, 2020. 110, it would be better not to put it in one paragraph format. Merge it with other sentences. A paragraph commonly consists of between 5 and 8 sentences.
Thank you for your observation and advice. Sentences of only one line have been merged with their corresponding paragraphs.
- had higher scores than those 208 who had bad mental health – what is the explanation for this finding?
We appreciate your comment as it is also applicable to the perception of physical health. The literature describes how quality psychiatric care leads to better health outcomes although, as ours is a cross-sectional study, it is not possible to determine whether high quality care produces better health in the patients treated or better health influences patient perception of the facility attended; a limitation already found in a similar study to ours. Future studies should examine in detail to what extent service quality improvements influence people's mental and physical health. These points are dealt with in the Discussion section, the paragraph referring to mental and physical health and in limitations.
- This finding is in line with previous studies that indicate that older people are usually less critical than younger ones when rating the quality of care received – what is the recommendation for this? Should differentiated services based on patients’ ages be provided?
Excellent questions. Thank you. Our recommendation in this respect would be that care should indeed be differentiated and provided in accordance with the person’s age, their individual needs and their environment. This recommendation has been added to the implications of the findings for clinical management at the end of the discussion.
- Was there an open-ended questionnaire in this study? If not, please consider suggesting further research to use the methods to gain recommendations from participants.
Thank you very much for this advice. The study did not include an open-ended questionnaire although it would without doubt contribute very valuable information in the form of suggestions from participants. This has been added as a recommendation at the end of the discussion alongside further research.
Thank you! Best of luck!
Reviewer 3 Report
The paper is in general well written. I have the following comments:
1. The recruited patients and staff were from community mental health facilities in the Barcelona area. It will be helpful to report the number and basic characteristics of these facilities to give a sense on how generalizable the study sample is to the whole country.
2. The study mainly relied on using instruments QPC-OP and QPC-OPS. It would be helpful if the authors provided more information on the concepts evaluated in these instruments. For example, what does "Encounter", "participation-empowerment", "participation-information"... really mean. Such information will help readers interpret the results and implications.
3. The Discussion section is a little too lengthy. The authors provided discussions on each component of survey findings and whether the finding align with previous literature. However, the authors should really discuss the implications of those findings to clinical management.
Author Response
The paper is in general well written. I have the following comments:
Dear Reviewer, thank you very much for the positive assessment of our manuscript. We have added a paragraph in the introduction according to a comment from another reviewer to improve the quality of the section. Below, we reply to your observations point by point:
- The recruited patients and staff were from community mental health facilities in the Barcelona area. It will be helpful to report the number and basic characteristics of these facilities to give a sense on how generalizable the study sample is to the whole country.
The community mental health facilities involved in our study belong to the same institution, which provides a specialized mental health service in agreement with the regional health system. Consequently, from the patient’s point of view, the service is public and free at the point of access. This information is included in section 2.2.
- The study mainly relied on using instruments QPC-OP and QPC-OPS. It would be helpful if the authors provided more information on the concepts evaluated in these instruments. For example, what does "Encounter", "participation-empowerment", "participation-information"... really mean. Such information will help readers interpret the results and implications.
Thank you. We appreciate the opportunity to describe the contents of the instruments in more detail. The corresponding paragraph in section 2.3 has been modified so that together with the name of the factor and the number of items, there is a brief explanation of the aspects covered in each factor.
- The Discussion section is a little too lengthy. The authors provided discussions on each component of survey findings and whether the finding align with previous literature. However, the authors should really discuss the implications of those findings to clinical management.
We are very grateful for your suggestions and apologize for the length of the discussion. We have revised it carefully although, with the addition of some sentences in line with reviewers’ recommendations, reduction of the number of words proved a little difficult. To close the discussion section, a paragraph has been added on the implications of the findings for clinical management.
Reviewer 4 Report
The method used to colect data, is described ,althoug could be represented using a process flow diagram that could improve to readers better understand the used method. Other issue is that could improve the paragraph "The main limitation is that as a comparative cross-sectional study it was not pos- 386 sible to assess any type of progress as follow-up assessments were not conducted. " to describe why is not be possible to conduct.
Author Response
The method used to colect data, is described ,althoug could be represented using a process flow diagram that could improve to readers better understand the used method.
Thank you for this recommendation. We have prepared a Flow Diagram of the process to aid readers as you suggest.
Other issue is that could improve the paragraph "The main limitation is that as a comparative cross-sectional study it was not pos- 386 sible to assess any type of progress as follow-up assessments were not conducted. " to describe why is not be possible to conduct.
The authors apologize for any confusion this sentence may have caused. We have modified it for clarification. Our intention was to explain that due to the cross-sectional nature of the study, no follow-up assessments were conducted and as a result we were not able to evaluate any progress.
Round 2
Reviewer 2 Report
Dear Authors,
Following are some points I recommend to improve:
1. Design - it sounds too brief to say just "Cross-sectional descriptive study." Explain what it is, and how it worked in your research.
2. the same institution in 10 areas in and around Barcelona. -- would be better to mention what is the institution name and did the areas mean (e.g. cities, locations, and in how many cities?) Explain why you would not mention the institution's name. If it is confidential, then explain why.
Author Response
Dear reviewer,
below we respond to your suggestions and observations
Dear Authors,
Following are some points I recommend to improve:
1. Design - it sounds too brief to say just "Cross-sectional descriptive study." Explain what it is, and how it worked in your research.
the word "comparative" was added to the name of the design for clarification, and a brief explanation of what this type of design consists of has subsequently been written.
2. the same institution in 10 areas in and around Barcelona. -- would be better to mention what is the institution name and did the areas mean (e.g. cities, locations, and in how many cities?) Explain why you would not mention the institution's name. If it is confidential, then explain why.
Thank you for giving us the opportunity to clarify this. We didn't put the name of the institution in the text as it appeared in the affiliations. Apologies for any confusion. There is no confidentiality issue. We have added it now and have given a better description of what we mean by 'areas'. We have added the following sentence to the text “Both samples were recruited from community mental health facilities from Parc Sanitari Sant Joan de Déu in 1 district from Barcelona city and 9 nearby towns”.
We would like to thank your contributions to the manuscript.